# Influence of Glycine Betaine (Natural and Synthetic) on Growth, Metabolism and Yield Production of Drought-Stressed Maize (*Zea mays* L.) Plants

**DOI:** 10.3390/plants10112540

**Published:** 2021-11-22

**Authors:** Sidra Shafiq, Nudrat Aisha Akram, Muhammad Ashraf, Pedro García-Caparrós, Omar M. Ali, Arafat Abdel Hamed Abdel Latef

**Affiliations:** 1Department of Botany, Government College University, Faisalabad 38000, Pakistan; sidrashafiq25@gmail.com; 2Department of Botany, University of Agriculture, Faisalabad 38000, Pakistan; ashrafbot@yahoo.com; 3Department of Agronomy, Polytechnic School, University of Almeria, 04120 Almeria, Spain; pedrogar123@hotmail.com; 4Department of Chemistry, Turabah University College, Turabah Branch, Taif University, Taif 21944, Saudi Arabia; om.ali@tu.edu.sa; 5Botany and Microbiology Department, Faculty of Science, South Valley University, Qena 83523, Egypt

**Keywords:** phytoextract, maize, biostimulants, water deficit, glycine betaine

## Abstract

A study was carried out to evaluate the effectiveness of sugar beet extract (SBE) and glycine betaine (GB) in mitigating the adverse effects of drought stress on two maize cultivars. Seeds (caryopses) of two maize cultivars, Sadaf (drought-tolerant) and Sultan (drought-sensitive) were sown in plastic pots. Plants were subjected to different (100%, 75% and 60% field capacity (FC)) water regimes. Then, different levels of SBE (3% and 4%) and GB (3.65 and 3.84 g/L) were applied as a foliar spray after 30 days of water deficit stress. Drought stress significantly decreased plant growth and yield attributes, chlorophyll pigments, while it increased relative membrane permeability (RMP), levels of osmolytes (GB and proline), malondialdehyde (MDA), total phenolics and ascorbic acid as well as the activities of superoxide dismutase (SOD) and peroxidase (POD) enzymes in both maize cultivars. Exogenous application via foliar spray with SBR or GB improved plant growth and yield attributes, chlorophyll pigments, osmolyte concentration, total phenolics, ascorbic acid and the activities of reactive oxygen species (ROS) scavenging enzymes (SOD, POD and catalase; CAT), but reduced leaf RMP and MDA concentration. The results obtained in this study exhibit the role of foliar-applied biostimulants (natural and synthetic compounds) in enhancing the growth and yield of maize cultivars by upregulating the oxidative defense system and osmoprotectant accumulation under water deficit conditions.

## 1. Introduction

Crop production and its distribution in the world are being considerably hampered due to drought stress as a consequence of the climatic change and anthropogenic activities [1,2]. In the coming years, the duration and frequency of drought periods will rise due to the current scenario of climate change resulting in one of the most important threats of the current century [3]. Under water scarcity, crop growth and yield reduce drastically and several physio-biochemical processes, such as nutrient metabolism, photosynthesis, uptake and translocation of ions, carbohydrate metabolism, respiration and chlorophyll biosynthesis, are adversely affected [4,5]. Moreover, an imbalance between generation and scavenging of reactive oxygen species (ROS) leads to damage of lipids, proteins, nucleic acids and, subsequently, sometimes plant cell death may take place [3].

In defense mechanisms, plants enhance the production of osmolytes or osmoprotectants under stress conditions including drought [6]. Osmoprotectants preserve the cellular apparatus from dehydration-induced injury without interfering with the normal metabolic processes at the cellular level. They comprise a wide variety of compounds/molecules, such as proline, glycine betaine (GB), trehalose, quaternary ammonium compounds, phenolics, sugars, etc. [7,8]. Glycine betaine is a quaternary ammonium compound endogenously synthesized in chloroplasts in response to abiotic stressors, such as drought and salinity [9,10]. Glycine betaine not only acts as an osmoregulator but also stabilizes the structures and activities of enzymes and protein complexes and maintains the integrity of membranes against the damaging effects of drought [11]. It has been shown that exogenous application of GB could be a promising way to directly maintain and enhance the growth and yield of maize [12,13,14]. Water deficit conditions alter the growth pattern of plants with inhibition in development both qualitatively and quantitatively [15,16]. Due to this fact, drought stress is closely associated with those factors which decrease the yield of plants [17,18]. Cereal crops employ a number of strategies for defense purposes at cellular and molecular levels against drought stress. They tend to adapt themselves according to external conditions. Plants change their metabolic pathways for their survival and synthesize special osmolytes to tolerate environmental stresses [19,20].

Maize (*Zea mays* L.), an important cereal crop, is reported to be sensitive to drought stress because its growth and yield are considerably affected by this stress [21]. For instance, it requires around 50–800 mm water to complete its life cycle that ranges from about 120–150 days [7]. Irrigation water is becoming a limiting factor for crops with time [22,23], but on the other hand, food demand is consistently increasing with a progressive increase in the human population [15,24]. At this stage, maize can be used as a supplement food to meet the challenges of food scarcity because of its higher yield than that of wheat and rice.

Plant biostimulants include diverse substances that enhance plant growth through the stimulation of natural processes such as nutrient uptake efficiency, and tolerance to abiotic stress conditions [25,26]. It has been reported that the application of natural biostimulants, such as plant extracts enriched with key biostimulants, can improve plant growth under stress conditions more than that by synthetic chemicals [27,28]. Sugar beet (*Beta vulgaris* L.) extract is reported to be an important source of sucrose and glycine betaine [29,30]. Sugar beet extract contains a variety of substances such as ascorbic acid, glycine betaine, vitamin E, sugars and amino acids, which are considered effective to offset stress-induced oxidative stress in plants [31]. For instance, Noman et al. [32] reported a positive impact of the application of sugar beet extract (SBE) as a biostimulant to ameliorate the adverse effects of drought on seed germination and growth of wheat plants. Nevertheless, the literature on the use of plant biostimulants, especially on the application of plant extracts, is very scarce, therefore, this study aimed to compare the effectiveness of sugar beet extract and the synthetic glycine betaine on maize plant growth and metabolism under water deficit conditions.

## 2. Results

Drought stress (75% and 60% field capacity (FC)) significantly (*p* ≤ 0.001) suppressed the growth (shoot and root fresh and dry biomass) as compared to the control (100% FC) of both maize cultivars. Foliage application with both natural and synthetic biostimulants, SBE and GB, respectively, considerably (*p* ≤ 0.001) boosted the weights of roots and shoots (both fresh and dry), particularly under water deficit conditions (Figure 1; Table 1). The response of cv. Sadaf was better in biomass production than that of cv. Sultan. Of all the exogenous treatments, sugar beet extract applied at the rate of 4% (40 g/L) was most effective in improving plant growth under drought stress conditions.

Drought stress caused a significant (*p* ≤ 0.001) reduction in leaf area per plant of both maize cultivars. Foliar application of SBE and GB considerably (*p* ≤ 0.001) improved the leaf area of both maize cultivars (Figure 1). Both maize cultivars showed a similar response to water stress as well as all of the different levels of GB applied externally. However, the interaction of drought stress, exogenous application and cultivars was statistically non-significant (Table 1).

Chlorophyll pigments (a and b and total chlorophyll) decreased markedly (*p* ≤ 0.01; Table 2) in both maize cultivars under water deficit stress (75% and 60% FC). Exogenous application (SBE and GB) prominently (*p* ≤ 0.01) increased the pigment concentration in both maize cultivars, Sadaf and Sultan (Figure 2). Sugar beet extract at the rate of 4% was the most useful in increasing pigment concentration in both cultivars. The analysis of variance (ANOVA) presented in Table 1 indicated that the interaction of drought stress, exogenous application and cultivars was non-significant in terms of chlorophyll pigments. Nevertheless, the chlorophyll a/b ratio remained unchanged under drought stress conditions and foliar application of both compounds. Both cultivars were similar in all chlorophyll-related attributes under different drought conditions. 

Drought induced a significant (*p* ≤ 0.001) increase in RMP of both maize cultivars. Exogenously applied varying levels of GB considerably increased the RMP of both maize cultivars. The effect of drought stress was relatively more prominent on the cultivar Sultan (Figure 3; Table 2). The interaction between drought stress and cultivars was significant (*p* ≤ 0.001), while among drought, treatments and cultivars was non-significant. 

Under drought stress conditions, a considerable (*p* ≤ 0.001; Figure 3) rise was observed in leaf proline and glycine betaine in both maize cultivars. The foliar application of SBE and GB also enhanced the levels of both leaf osmolytes (proline and glycine betaine) under water deficit conditions. The drought-tolerant (DT) cultivar (Sadaf) showed higher osmolytes’ concentration than that in the drought-sensitive (DS) cultivar (Sultan) (Figure 3; Table 2). For both osmolytes, there was a non-significant interaction between drought stress × cultivars, drought stress × treatments, treatments × cultivars and drought stress × treatments × cultivars. 

Drought stress induced a marked (*p* ≤ 0.001) increase in leaf total phenolics and ascorbic acid concentrations in both maize cultivars. Exogenously applied SBE and GB also elevated leaf total phenolics and ascorbic acid concentrations in both maize cultivars under water scarcity. The response of cv. Sadaf was better than that of cv. Sultan under water stress conditions (Table 2; Figure 4).

Water deficit conditions markedly (*p* ≤ 0.01) enhanced the accumulation of malondialdehyde (MDA) in both maize cultivars. However, exogenous application of SBE and GB considerably (*p* ≤ 0.01; Figure 5; Table 2) decreased the MDA concentration in both maize cultivars under water deficit regimes. The cultivar Sadaf showed a higher MDA concentration than that of cv. Sultan under different water regimes. Likewise, a significant interaction between drought × cultivars was only observed for this attribute.

Hydrogen peroxide concentration in both maize cultivars increased significantly (*p* ≤ 0.001) under water deficit regimes. However, foliar spray with both compounds (GB and SBE) did not alter the leaf H_2_O_2_ concentration under water deficit conditions. The DS cultivar (Sultan) showed higher leaf H_2_O_2_ concentration than the DT cultivar (Sadaf) under water deficit conditions (Figure 5; Table 2).

Under drought stress conditions, a considerable (*p* ≤ 0.001) increase was observed in the activities of superoxide dismutase (SOD) and peroxidase (POD) enzymes, whereas the activity of catalase (CAT) remained unchanged under water deficit conditions. Foliar application of SBE and GB enhanced the activities of CAT, SOD and POD under the control as well as water stress conditions. Sugar beet extract (4%) and GB at higher concentrations were most effective in improving the activities of antioxidant enzymes. The response of both maize cultivars was almost similar in the activities of enzymatic antioxidants (Figure 6).

All yield attributes (number of grains per cob, 100-grain weight and grain yield plant) declined significantly (*p* < 0.001) under water deficit conditions. Exogenous application of SBE and GB considerably enhanced the yield attributes in both maize cultivars under drought stress. Cultivar Sadaf showed higher yield attributes than did cultivar Sultan under water deficit conditions (Figure 7; Table 3).

The correlation coefficient (r^2^) data showed that shoot fresh weight was positively correlated with other growth attributes, such as root fresh and dry weights (r^2^ = 0.745 *) and shoot dry weight (r^2^ = 0.776 *), while negatively (r^2^ = −0.613 *) linked to relative membrane permeability. Shoot dry weight was correlated with root fresh and dry biomass (r^2^ = 0.715 * and 0.716 *, respectively). Root fresh weight was positively related to root dry weight (r^2^ = 0.999 ***) and root dry weight was negatively associated with RMP (r^2^ = −0.616 *). All of the growth attributes were also positively correlated to yield attributes, such as grain yield per plant (r^2^ = 0.784 *) and number of grains per cob (r^2^ = 0.687 *). A positive correlation of chlorophyll a with the chl. a/b ratio (r^2^ = 0.672 *), chlorophyll b and total chlorophyll (r^2^ = 0.972 ***) was also noted. Leaf proline was positively correlated with total phenolics (r^2^ = 0.62 *) and the activity of the SOD enzyme (r^2^ = 0.608 *). Grain yield per plant was negatively associated with relative membrane permeability and positively correlated with leaf area (r^2^ = 0.808 **). All of the growth attributes were found to be interconnected with each other.

## 3. Discussion

Abiotic stresses suppressed the growth of different plants, including maize plants [33,34,35]. In the present study, plant growth and chlorophyll pigments declined in both maize cultivars under both drought stress regimes. The reduction in photosynthetic pigments under stress conditions such as drought may be due to impaired biosynthesis or breakdown of chlorophyll pigments and related compounds [36]. The foliar application of SBE and GB improved chlorophyll pigments in both maize cultivars under water deficit conditions. Likewise, earlier in rapeseed, foliar-applied GB was reported to improve chlorophyll concentration under drought stress conditions [36]. This enhancement in pigment concentrations may have been due to the role of GB in protecting the photosynthetic apparatus and stabilizing the structures of Rubisco and membranes in plants under water deficit conditions [37,38]. Moreover, it is known that GB improves the efficiency of photosynthetic machinery [38,39].

Water deficiency also reduces leaf area per plant as it can induce impairment in water relations and gas exchange characteristics [24]. In this experiment, leaf area decreased in both maize cultivars under water deficit conditions. Similar results have been reported under drought conditions in other crops such as amaranth [40], quinoa [41] and tobacco [42]. Moreover, exogenously supplemented SBE and GB improved leaf area in both water-starved maize cultivars. A GB-induced increase in leaf area in water-stressed plants has been reported in tomato [43] and rice [44], and it was ascribed to the ameliorative effect of GB on photosynthetic processes in stressed plants [45].

Alteration in membranes is often linked to the rise in their permeability and loss of integrity [46,47]. Changes in the permeability of membranes may occur due to leakage from cells occurring due to damage to the components of the membrane in the lipid matrix [48]. In the present study, the relative membrane permeability increased in the maize cultivars subjected to drought stress conditions, as well as by the foliar application of SBE as well as GB. Li-Ping et al. [49] also reported that drought stress enhanced RMP in maize plants, whereas Ahmed et al. [50] reported that foliar spray of GB improved the membrane stability in drought-stressed wheat plants as observed in the present study. 

Organic osmolytes, such as proline and GB, are known to maintain the water potential of cells, protect macromolecules and enzymes from oxidative damage, increase activities of enzymes, reduce the concentration of H_2_O_2_ and improve the tolerance of plants against oxidative stress conditions [51,52,53]. In the current experiment, both maize cultivars showed increased levels of leaf proline and glycine betaine under drought stress conditions. Generally, a high accumulation of GB or proline is considered as a prospective indicator of stress tolerance [9,10,53]. Moreover, in this experiment, foliar application of SBE and GB increased the concentration of these osmolytes in both maize cultivars under drought conditions. Glycine betaine applied as a foliar spray has been reported to enhance the endogenous levels of both GB and proline in many plant species [54,55], suggesting the positive role of this chemical compound in enhancing drought stress tolerance by upregulating the mechanisms involved in growth and yield production under stress conditions. 

In this experiment, lipid peroxidation appraised in terms of MDA concentration and H_2_O_2_ concentration in both maize cultivars increased under drought conditions, while both parameters decreased by the foliar spray of SBE and GB. The protective role of GB in reducing lipid peroxidation and H_2_O_2_ concentration under drought conditions has been reported in different crops such as rice [56], safflower [10] and rapeseed [36]. Concerning the role of SBE, Noman et al. [32] also reported a decrease in MDA and H_2_O_2_ concentrations due to foliar application of SBE on wheat plants under water deficit conditions. They suggested that the decreasing trend in these biomolecules be due to the increase in the activities of antioxidant enzymes involved in scavenging ROS under water stress conditions. 

Non-enzymatic antioxidants, such as total phenolics, are secondary metabolites that are believed to be involved in the prevention of lipid peroxidation and denaturation of proteins, scavenging of ROS and preventing DNA damage [57]. In this experiment, leaf total phenolics concentration increased under water scarcity in both maize cultivars, and the exogenous application of GB and SBE also increased total phenolics. The foliar application of GB has already been reported to enhance total phenolics under drought conditions in cotton [58] and wheat [59]. Ascorbic acid is a key antioxidant that plays an important role in drought stress tolerance and it improves plant growth and production [60]. Increased ascorbic acid (AsA) concentrations under stress conditions can prevent oxidative damage by reducing the production of ROS [61,62]. Leaf ascorbic acid concentration in both maize cultivars (Sadaf and Sultan) increased by drought as well as by foliar application of SBE and GB both under the control and water deficit conditions. Similarly, foliar application of GB increased AsA concentrations in water-stressed plants of bread wheat [63] and strawberry [64].

Disruption in cell homeostasis caused by ROS is believed to be alleviated by the action of various enzymes such as CAT, SOD and POD [65,66,67]. However, tolerance of plants against abiotic stresses has been linked to the mechanism of ROS generation and scavenging by antioxidative capacity [5]. In our investigation, the activities of enzymatic antioxidants (POD and SOD) increased under drought stress conditions in both maize cultivars. Similar findings have been reported in faba bean [68], and Vicia faba [69] showing decreased oxidative stress caused by ROS generation. The exogenous application of SBE and GB also enhanced the antioxidant activity in both maize cultivars. So, improvement in plant growth of maize plants can be interlinked with up-regulation of SOD and POD enzymes. An enhancement in the antioxidant activity was reported by Farooq et al. [56] under foliar application of GB in rice grown under water deficit conditions. Similarly, in oat plants, Shehzadi et al. [55] reported that foliar application of GB improved antioxidants activities resulting in improved drought stress tolerance.

In the present study, yield attributes decreased significantly under water deficit conditions in both cultivars. Previous reports show that plant growth and, consequently, the yield of maize were severely affected by drought stress [70,71]. However, the foliar application of SBE and GB resulted in enhanced yield attributes in cvs. Sultan and Sadaf. Likewise, Raza et al. [59] reported that the yield of wheat was improved by the application of GB under drought stress conditions. Exogenous application of GB at the vegetative stage may improve translocation of assimilates and water relation attributes leading to increased grain yield of maize plants in the present study. Moreover, the increase in grain yield may have been due to the increased production of endogenous GB and essential amino acids [72]. It was also reported that foliar application of SBE caused an increase in plant biomass of wheat plants which was found to be associated with an increased antioxidative defense system [32].

## 4. Materials and Methods

### 4.1. Plant Material and Growth Conditions

A pot experiment was arranged in a completely randomized design from July to October 2019 at the research area of Government College University Faisalabad, Punjab, Pakistan. For this study, ten seeds of two maize cultivars, Sadaf (drought-tolerant; DT) and Sultan (drought-sensitive; DS) were sown in plastic pots (height, 42.5 cm and diameter, 23 cm) each filled with 8 kg sandy loam soil. After one week, five uniform seedlings were maintained in each plastic pot. The seedlings were allowed to grow for 25 days under environmental conditions (average temperature, day (36.55 °C) and night (25.85 °C), average relative humidity, 68.4% and average sunshine 7.9 h). Thereafter, the plants were subjected to varying levels (100%, 75% and 60% FC) of water stress with four replicates. After 30 days of drought stress treatments, sugar beet extract (SBE) at the rate of 3% and 4% along with synthetic glycine betaine (GB; MP Biomedical Inc.) at the rate of 3.65 g/L (0.365%) and 3.84 g/L (0.384%) were applied as a foliar spray (10 mL per plant) using a manual plastic sharped nasal sprayer. Sugar beet was obtained from the local market of Faisalabad, Pakistan and a stock solution (4%) was prepared by using an electric grinder. Then, the stock solution was used to prepare another concentration of sugar beet, and the GB contents in SBE were determined. These GB levels were calculated based on the GB present in 3% (30 g/L) and 4% (40 g/L) SBE, respectively. The GB contents in the SBE were determined following Grieve and Grattan [73]. The plants were harvested after two weeks of foliage treatments to analyze the following attributes.

### 4.2. Plant Fresh and Dry Biomass

Two plants from each replicate were harvested and the shoots and roots were separated. The fresh weights of both plant parts were noted using an analytical balance with 98.5% efficiency. The shoots and roots were air-dried before being placed in an oven at 70 °C for 3 days and then having their dry weights recorded.

### 4.3. Leaf Area per Plant

Leaf area was measured according to the following formula:Leaf area per plant (cm^2^) = length × width × number of leaves × correction factor

### 4.4. Chlorophyll Pigments

The fresh leaf sample (each sample 0.5 g) was homogenized in 10 mL of acetone (80%; *v*/*v*). The extract was filtered, and the filtrate was kept at 4 °C for 24 h. Then, the optical density (OD) of the supernatant was determined spectrophotometrically at 663, 645 and 480 nm following Arnon [74].

### 4.5. Relative Membrane Permeability (RMP)

Fresh leaf sample (each 0.5 g) was chopped into small uniform pieces and placed in 10 mL of deionized water in a test tube for one night for determining the electrical conductivity (EC0). Then, autoclaved for one hour and measured the EC1, followed by being kept overnight at 4 °C and the EC2 recorded to calculate the RMP following Yang et al. [75].

### 4.6. Leaf Proline

Leaf proline concentration was determined according to the method of Bates et al. [76]. The fresh leaf sample (0.5 g) was homogenized in 10 mL of 3% (*w*/*v*) sulfosalicylic acid and then centrifuged at 10,000× *g* for 10 min at 4 °C. To the supernatant (2 mL), acid ninhydrin (2 mL) and glacial acetic acid (2 mL) were added. Then, the mixture was incubated for 30 min at 100 °C in a water bath. After cooling, the mixture was extracted with toluene and the absorbance of the upper layer was recorded spectrophotometrically at 520 nm.

### 4.7. Leaf Glycine Betaine

The leaf sample (0.5 g) was triturated in 10 mL toluene (0.5%). After filtration, 1 mL of the extract was mixed with 1 mL of 2 N H_2_SO_4_ solution. Then, 0.5 mL of the mixture was added together with 0.2 mL of potassium tri-iodide (KI_3_) in a test tube and all of the samples were placed in an ice bath for 90 min and then 5 mL of 1,2 dichloroethane and 2.8 mL of distilled water were added to each test tube. By passing a continuous stream of air for 1–2 min, two layers appeared. The upper aqueous layer was discarded, and the optical density of the lower organic layer was recorded at 365 nm using a spectrophotometer.

### 4.8. Malondialdehyde (MDA)

The leaf peroxidation in the form of malondialdehyde concentration was determined following the protocol of Cakmak and Horst [77]. The fresh leaf sample (0.25 g) was homogenized in 3 mL of 1% (*w*/*v*) trichloroacetic acid (TCA). The mixture was centrifuged at 10,000× *g* for 15 min. An aliquot (1 mL) was treated with 4 mL of 0.5% thiobarbituric acid (TBA), which was prepared in 20% TCA. After that, this mixture was boiled at 95 °C for 30 min and cooled in an ice bath. The absorbance of the treated mixture was recorded at 532 and 600 nm using a spectrophotometer.

### 4.9. Hydrogen Peroxide (H_2_O_2_)

The hydrogen peroxide (H_2_O_2_) concentration was determined in maize plant leaves according to the protocol reported by Velikova et al. [78]. The fresh leaf sample (0.25 g) was homogenized in 5 mL of 0.1% (*w*/*v*) trichloroacetic acid (TCA) using a mortar and pestle. After filtering the extract, an aliquot (0.5 mL) was taken in a test tube, and 1 mL of potassium iodide and 0.5 mL of phosphate buffer were added to it. The test tubes were kept at room temperature for 20 min, and the absorbance was recorded at 390 nm using a spectrophotometer.

### 4.10. Total Phenolics

The total phenolics concentration was determined in the leaves of maize plants according to the method of Julkenen-Titto [79]. Fresh leaf samples (each 0.1 g) were extracted, each in 5 mL of 80% acetone, and then all of the extracts were centrifuged. To 0.1 mL extract, 2 mL of deionized water and 1 mL of Folin–Ciocalteau’s reagent were added and the mixture was shaken well. Then, 5 mL of 20% sodium carbonate (Na_2_CO_3_) were added to the mixture and brought the final volume to 10 mL by adding distilled water. The amount of total phenolics was estimated by reading the treated samples at 750 nm using a spectrophotometer.

### 4.11. Ascorbic Acid (AsA)

According to the method proposed by Mukherjee and Choudhuri [80], fresh leaf samples (0.25 g) were homogenized each in 10 mL of 6% TCA and the extract was filtered. To 4 mL of the extract contained in a test tube, 2 mL of dinitrophenyl hydrazine (2% in 9 N H_2_SO_4_) and one drop of 10% thiourea (prepared in 70% ethanol) were added. The mixture was then boiled for 15 min and cooled at room temperature. Then, 5 mL of 80% (*v*/*v*) H_2_SO_4_ was added to the mixture. The absorbance of the samples was recorded at 530 nm spectrophotometrically.

### 4.12. Activities of Enzymatic Antioxidants

Phosphate buffer (10 mL, pH 7.8) was used for the extraction of antioxidant enzymes from fresh leaf samples (each 0.5 g). The extract was centrifuged, and the aliquot was used to appraise the activities of different enzymes (catalase (CAT), peroxidase (POD) and superoxide dismutase (SOD)). The activity of SOD enzyme was estimated following the protocol reported by Giannopolitis and Ries [81]. However, the activities of CAT and POD enzymes were determined according to the method of Chance and Maehly [82].

### 4.13. Yield Attributes

At the maturity stage, the cobs were harvested and yield attributes, such as 100-grain weight, number of grains per cob and grain yield per plant, were determined.

### 4.14. Statistical Analysis

A completely randomized design with four replicates was employed and the data obtained were analyzed for working out a three-way analysis of variance (ANOVA) using the statistical software Costat, version 6.303. A correlation analysis among all the above-mentioned variables was also worked out.

## 5. Conclusions and Recommendations

The results of this experiment showed that water-stressed maize plants showed a reduction in plant biomass and yield attributes, and leaf area and pigment concentrations, while they exhibited an increase in relative membrane permeability, levels of leaf proline, glycine betaine, MDA, H_2_O_2,_ total phenolics, ascorbic acid and the activities of SOD and POD enzymes. The drought-tolerant cultivar (Sadaf) showed better performance than the drought-sensitive cultivar (Sultan) under drought conditions. The foliar application of SBE and GB enhanced plant biomass and yield attributes, leaf area, leaf pigment concentration, the levels of RMP, leaf proline, glycine betaine, total phenolics, ascorbic acid and the activities of the antioxidant enzymes, while it reduced lipid peroxidation. Both chemicals (SBE and GB) were effective in improving plant growth and the oxidative defense system by accumulating high amounts of proline, GB, and phenolics, and due to enhanced activities of key antioxidant enzymes in both maize cultivars. These results suggested that rather than using synthetic chemicals or growth regulators, natural chemicals such as SBE should be widely used to ameliorate the adverse effect of drought stress on various crops. SBE is a cheap source of various nutrients that can be helpful in ameliorating the adversities of drought stress conditions. It can be recommended for future research on different crop plants to improve the growth and production under different stress conditions.

## Figures and Tables

**Figure 1 plants-10-02540-f001:**
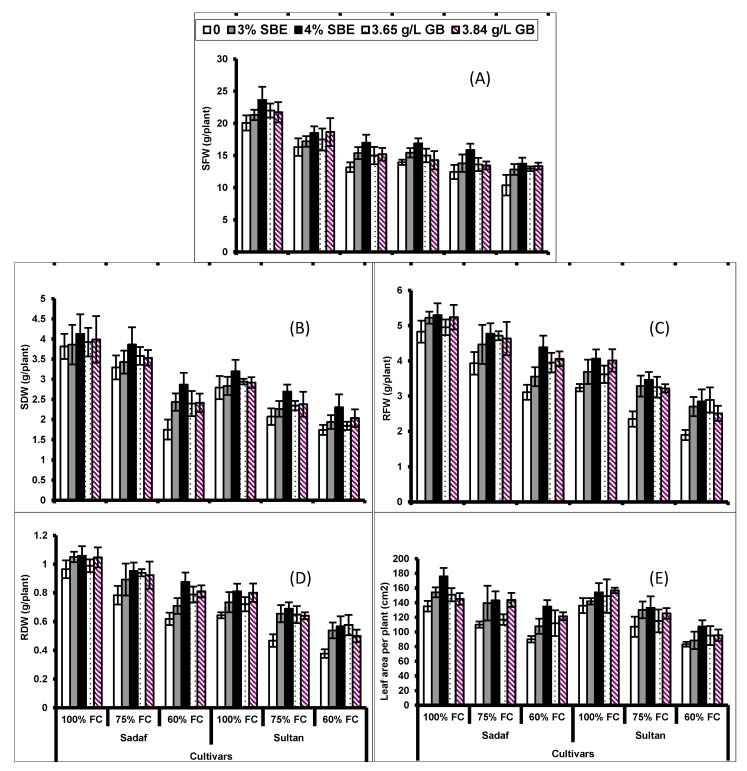
(**A**) Shoot fresh weight (SFW), (**B**) shoot dry weight (SDW), (**C**) root fresh weight (RFW), (**D**) root dry weight (RDW) and (**E**) leaf area of two maize cultivars foliar-fed with different levels of sugar beet extract (SBE) and glycine betaine under water deficit conditions (Mean ± S.E.).

**Figure 2 plants-10-02540-f002:**
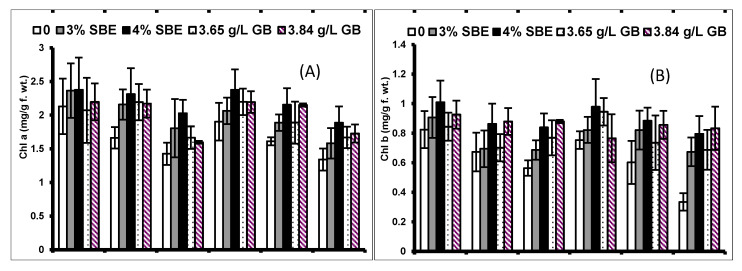
(**A**) Chlorophyll a (Chl a), (**B**) chlorophyll b (Chl b), (**C**) total chlorophyll and (**D**) chlorophyll a/b of two maize cultivars foliar-fed with varying levels of sugar beet extract (SBE) and glycine betaine under water deficit conditions (Mean ± S.E.).

**Figure 3 plants-10-02540-f003:**
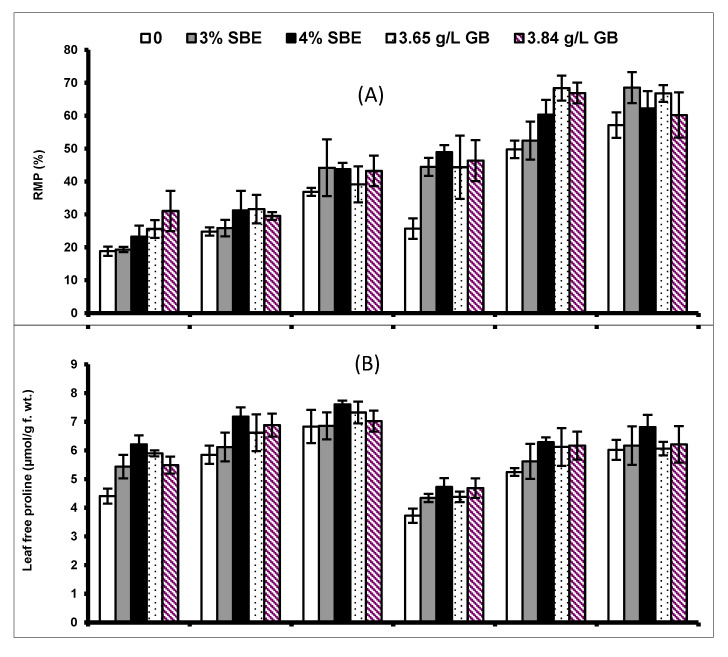
(**A**) Relative membrane permeability (RMP), (**B**) leaf free proline and (**C**) glycine betaine (GB) of two maize cultivars foliar-fed with varying levels of sugar beet extract (SBE) and glycine betaine under water deficit conditions (Mean ± S.E.).

**Figure 4 plants-10-02540-f004:**
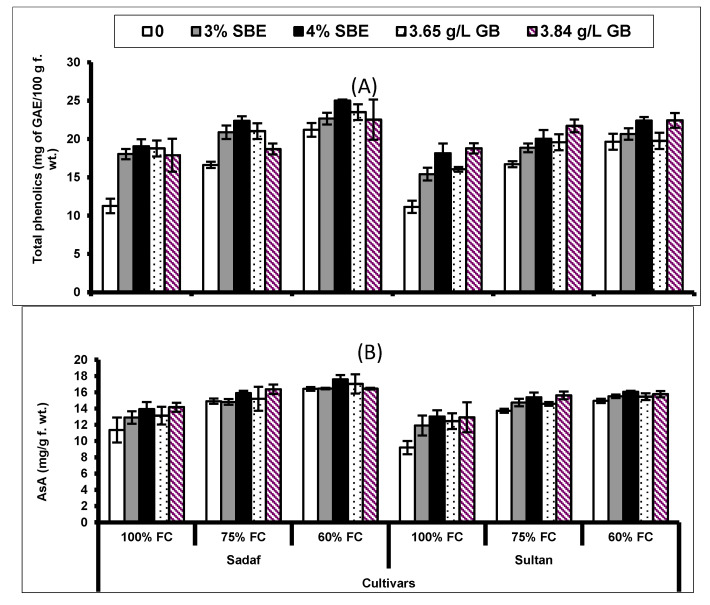
(**A**) Total phenolics and (**B**) ascorbic acid (AsA) of two maize cultivars foliar-fed with varying levels of sugar beet extract (SBE) and glycine betaine under water deficit conditions (Mean ± S.E.).

**Figure 5 plants-10-02540-f005:**
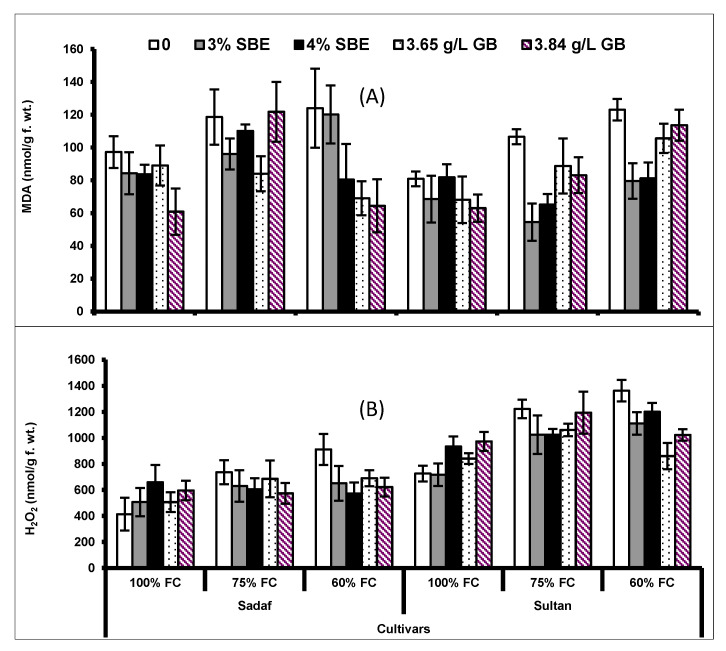
(**A**) Malondialdehyde (MDA) and (**B**) hydrogen peroxide (H_2_O_2_) of two maize cultivars foliar-fed with varying levels of sugar beet extract (SBE) and glycine betaine under water deficit conditions (Mean ± S.E.).

**Figure 6 plants-10-02540-f006:**
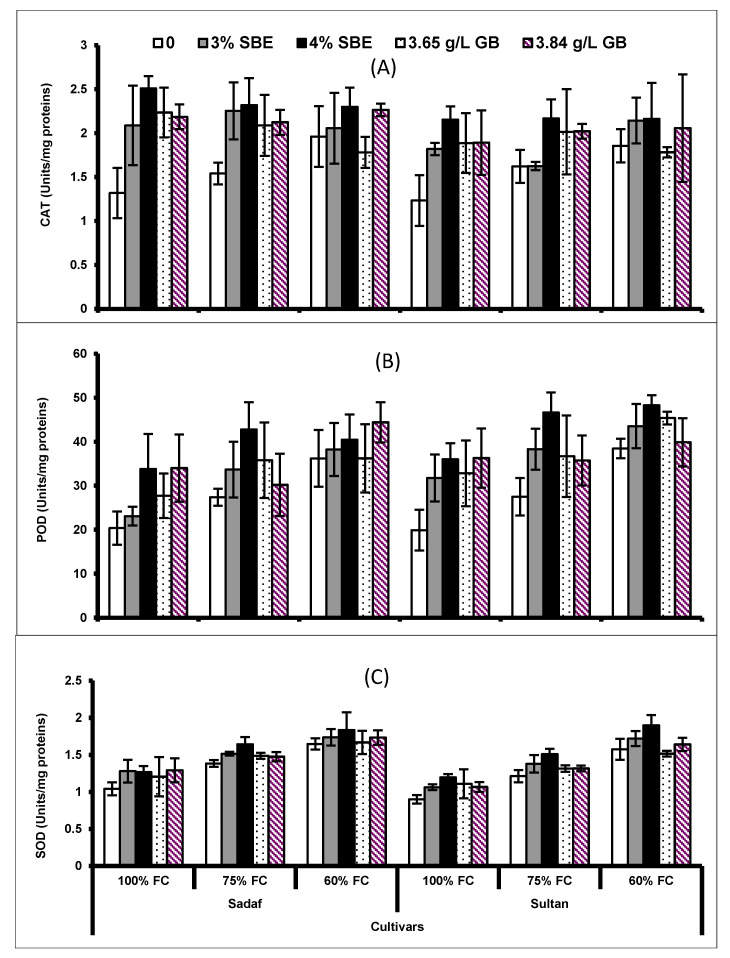
(**A**) Catalase (CAT), (**B**) peroxidase (POD) and (**C**) superoxide dismutase (SOD) of two maize cultivars foliar-fed with varying levels of sugar beet extract (SBE) and glycine betaine under water deficit conditions (Mean ± S.E.).

**Figure 7 plants-10-02540-f007:**
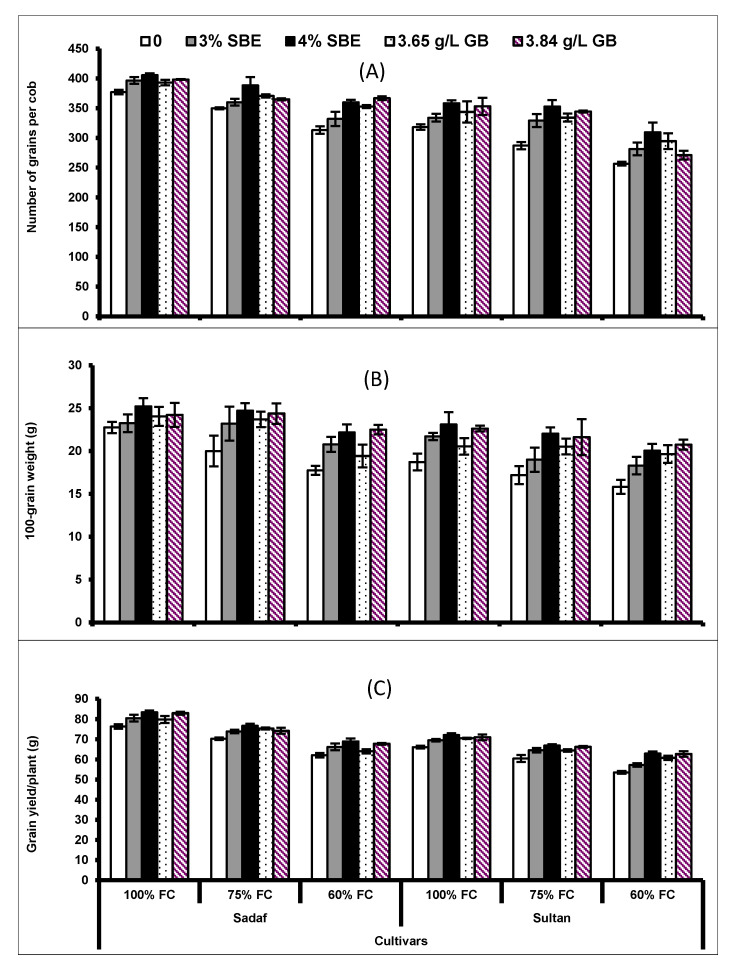
(**A**) Number of grains, (**B**) 100-grain weight and (**C**) grain yield of two maize cultivars foliar-fed with varying levels of sugar beet extract (SBE) and glycine betaine under water deficit conditions (Mean ± S.E.).

**Table 1 plants-10-02540-t001:** Analyses of variance data for growth attributes of maize cultivars subjected to varying levels of sugar beet extract (SBE) and glycine betaine under water deficit conditions.

Source of Variation	df	Shoot FW	Shoot DW	Root FW	Root DW
Drought stress (D)	2	207.4 ***	16.32 ***	15.08 ***	0.605 ***
Treatments (Trs)	4	29.42 ***	1.110 *	2.855 ***	0.115 ***
Cultivar (Cvs)	1	543.3 ***	22.45 ***	53.83 ***	2.165 ***
D x Trs	8	0.983 ^ns^	0.072 ^ns^	0.164 ^ns^	0.006 ^ns^
D x Cvs	2	45.01 ***	1.707 **	0.072 ^ns^	0.003 ^ns^
Cvs x Trs	4	0.777 ^ns^	0.026 ^ns^	0.065 ^ns^	0.002 ^ns^
D x Trs x Cvs	8	1.139 ^ns^	0.044 ^ns^	0.111 ^ns^	0.004 ^ns^
Error	90	5.613	0.030	0.337	0.013

^ns^ = non-significant; *, ** and *** = significant at 0.05, 0.01 and 0.001 levels, respectively.

**Table 2 plants-10-02540-t002:** Analyses of variance data for various physio-biochemical attributes of maize (*Zea mays* L.) cultivars subjected to varying levels of sugar beet extract (SBE) and glycine betaine under water deficit conditions.

Source of Variation	df	Chl a	Chl b	Total chl	Chl a/b Ratio	Leaf Area	RMP	Proline	GB
Drought stress (D)	2	0.298 **	2.748 ***	4.757 ***	0.018 ^ns^	21,451.8 ***	3804.7 ***	33.16 ***	4059.5 ***
Treatments (Trs)	4	0.257 **	0.803 *	1.936 ***	0.012 ^ns^	3124.8 ***	475.7 ***	4.218 ***	786.4 **
Cultivar (Cvs)	1	0.043 ^ns^	0.308 ^ns^	0.583 ^ns^	0.019 ^ns^	3519.2 *	16,742.4 ***	23.10 ***	7586.4 ***
D x Trs	8	0.036 ^ns^	0.044 ^ns^	0.091 ^ns^	0.018 ^ns^	224.2 ^ns^	95.76 ^ns^	0.332 ^ns^	106.1 ^ns^
D x Cvs	2	0.025 ^ns^	0.027 ^ns^	7.141 ^ns^	0.017 ^ns^	564.9 ^ns^	428.1 **	0.565 ^ns^	340.4 ^ns^
Cvs x Trs	4	0.021 ^ns^	0.071 ^ns^	0.066	0.008 ^ns^	251.1 ^ns^	90.01 ^ns^	0.201 ^ns^	109.8 ^ns^
D x Trs x Cvs	8	0.010 ^ns^	0.029 ^ns^	0.055 ^ns^	0.005 ^ns^	150.7 ^ns^	67.95 ^ns^	0.123 ^ns^	35.21 ^ns^
Error	90	0.054	0.283	0.327	0.027	524.07	81.29	0.657	189.2
	**df**	**AsA**	**MDA**	**H_2_O_2_**	**Total phenolics**	**CAT**	**POD**	**SOD**	
Drought stress (D)	2	142.9 ***	3845.8 **	543,596.6 ***	307.5 ***	0.109 ^ns^	1329.7 ***	3.083 ***	
Treatments (Trs)	4	13.90 ***	2849.4 **	61,664.1 ^ns^	90.48 ***	1.493**	527.4 **	0.224 **	
Cultivar (Cvs)	1	31.13 ***	2615.7 *	4,659,542.8 ***	43.95 **	0.886 ^ns^	373.3 ^ns^	0.428 **	
D x Trs	8	2.677 ^ns^	966.6 ^ns^	105,882.9 **	9.463 *	0.252 ^ns^	73.12 ^ns^	0.017 ^ns^	
D x Cvs	2	1.158 ^ns^	3158.9 **	67,809.4 ^ns^	5.397 ^ns^	0.097 ^ns^	2.494 ^ns^	0.032 ^ns^	
Cvs x Trs	4	0.747 ^ns^	1537.5 ^ns^	29,364.8 ^ns^	15.37 **	0.047 ^ns^	37.59 ^ns^	0.011 ^ns^	
D x Trs x Cvs	8	0.294 ^ns^	1058.06 ^ns^	26,489.2 ^ns^	1.461 ^ns^	0.074 ^ns^	28.65 ^ns^	0.006 ^ns^	
Error	90	2.430	637.7	36,554.2	4.126	0.329	129.9	0.053	

ns = non-significant; *, ** and *** = significant at 0.05, 0.01 and 0.001 levels, respectively.

**Table 3 plants-10-02540-t003:** Analyses of variance data for yield attributes of maize (*Zea mays* L.) cultivars subjected to varying levels of sugar beet extract (SBE) and glycine betaine under water deficit conditions.

Source of Variation	df	No of Grains/Plant	100-Grain Weight	Grain Yield/Plant
Drought stress (D)	2	29,868.1 ***	87.01 ***	1584.7 ***
Treatments (Trs)	4	6866.4 ***	67.21 ***	172.6 ***
Cultivar (Cvs)	1	77,216.1 ***	177.3 ***	2377.1 ***
D × Trs	8	155.5 ^ns^	1.935 ^ns^	3.097 ^ns^
D × Cvs	2	1593.9 **	5.861 ^ns^	50.46 ***
Cvs × Trs	4	209.08 ^ns^	0.873 ^ns^	3.964 ^ns^
D × Trs × Cvs	8	561.4 ^ns^	2.374 ^ns^	5.800 ^ns^
Error	90	279.08 ^ns^	4.875 ^ns^	4.604

^ns^ = non-significant; ** and *** = significant at 0.01 and 0.001 levels, respectively.

## Data Availability

The datasets generated and/or analyzed during the current study are available from the corresponding author upon reasonable request.

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
