# Peer review of "Influence of Glycine Betaine (Natural and Synthetic) on Growth, Metabolism and Yield Production of Drought-Stressed Maize (Zea mays L.) Plants"

_plants, 2021, doi:10.3390/plants10112540_

Round 1

Reviewer 1 Report

Dear Authors

The manuscript titled ‘Influence of Glycinebetaine (Natural and Synthetic) on Growth, Metabolites and Yield Production of Drought-Stressed Maize (Zea mays L.) Plants' is written clearly and concisely. However, I have some comments that should be taken into account by the Authors when revising this manuscript.

  1. SI Units(International System of Units) should be used.
  2. Keywords ‘water deficit conditions’ – remove ‘conditions’.
  3. There are too many references throughout the manuscript. Limit self-citations to 2-3. In the Introduction, use single references instead of several to support a single scientific statement (e.g. lines 48; 62; 64; 70; 72; 74). Avoid references to other plant species, especially in relation to morphological features (e.g. lines 280-285). References to species unrelated to maize are allowed only for physiological characteristics.
  4. In the Materials and Methods chapter, the authors state: ', ... seeds of two maize cultivars, ... were sown in plastic pots ... Germination percentage was noted after one week ..... The seedlings were allowed to grow for 25 days under natural environmental conditions ... Thereafter, the plants were subjected to varying levels .... of water stress .... After 30 days of drought stress treatments, sugar beet extract ... along with synthetic glycinebetaine ... were applied as a foliar spray ... ' It follows that neither the water stress nor the application of SBE and GB could affect the germination capacity described in the Results (lines 84-88) and Discussion (lines 257-266) sections.
  5. The description of the RMP response (lines 109-111; 290-291) is not fully understood and seems inconsistent with the results presented in Figure 3 A.
  6. Figure 1 -Data on the experimental factors on the OX axis are missing.
  7. The title of Table 1 is imprecise. What exactly is presented in this table. Moreover, the layout of this table is unacceptable. I propose to divide it into 3-4. Individual tables should be placed in the text in the order in which they are described.
  8. Table 2 is unreadable. I propose to include significant values of the correlation coefficients in the text instead of in the table.
  9. In the Materials and Methods section, the Authors did not provide how the leaf area was measured.
  10. The description of the SBE and GB applications is inaccurate. It is not known what doses were applied per plant or per pot.
  11. The number of grains should be provided as the number per cob instead of the number per plant.

Reviewer 2 Report

Manuscript ID: plants-1407037

Influence of Glycinebetaine (Natural and Synthetic) on Growth, Metabolites and Yield Production of Drought-Stressed Maize (Zea mays L.) Plants

The manuscript describes results of pot experiment on effect of sugar beet extract and the synthetic glycine betaine on maize growth and metabolism under water deficit conditions. The subject is actual and results can provide an advance in current knowledge to using natural chemicals as biostimulants to improve plant growth under abiotic stress conditions, however the results are not clearly presented. In this form, the scientific soundness of the manuscript is low. The manuscript can be accepted for publishing in Plants after major revision.

The hypothesis was made in the study should be clearly stated.

The section Materials and Methods important information are missing. The experimental design should be more detail described. Authors mentioned that" the data obtained were analyzed for working out a three-way analysis of variance (ANOVA)" without specifying what were the experimental factors. Water regimes was the first or the second factor? Levels of SBE and GB was another factor? How cultivar was incorporated into the design? It is not clear. What was the control treatment? It should be clearly stated. How the leaf area was determined?

Results of the experiment are improperly presented. Effects of the main factors and it significant interactions should be clearly presented according to the ANOVA. Authors need to first present the ANOVA table for everything that was tested, and then describe results according to the ANOVA. The effects of interaction D × Trs × Cvs presented in Figures 1-7 are not statistically confirmed (please see Table 1). The Authors statements  should be consistent with the data presented in the figures or tables.

Lines 84-89 should be removed. Drought stress or SBE and GB application could not affect maize germination due to plants were grown for 25 days under natural environmental conditions. Thereafter, the plants were subjected to varying levels of water stress and treatment with SBE and GB.

Results of the study should be clearly compared with a previous studies. Authors should better discuss significance of their findings.

Conclusions should be supported by the results. Recommendations for the future experiments which can be based on your results should be added.

Reviewer 3 Report

In my opinion, the experiment should be repeated in order to get the reported positive results twice.

There are a lot of formal mistakes or bad editing, above all those regarding the figures and tables, please find them below:

Line 2 and throughout the whole manuscript: rewrite glycinebetaine for glycine betaine.

Line 3: Is “Metabolites” the right word or maybe “Metabolism”.

Line 18: Please don´t add a period after the letters “F” and “C” of Field capacity (better: “FC” than “F.C.”)

Line 61: Add the scientific name for maize, just once in the first apparition in the introduction.

Line 63: There are maize cycles much longer than 110 days.

Line 73: Italics for Beta vulgaris.

Lines 84-89: It is impossible to see the differences between maize cultivars and water regimes in figure 1 as there is a big lack of information in the graphs (varieties and water regimes).

Line 165: Table 1 should be splitted at least into 2 tables, one for agronomic assessments and other for biochemical assessments, with vertical lines erased.

Figures 1 and 2 are very small and Table 2 is impossible to read, at least for me.

Lines 345-348: I don´t know if the improvement in grain yield observed when treating with both products is enough to justify the use of the products as authors use a strange yield unit, weight of grains per plant instead of kg/ha. Can the authors translate their units in a more used yield unit and compare it with bibliography?

Line 359: Was the experiment conducted under greenhouse or growing chamber conditions?

Line 362: dimensions of the pots? Height and diameter?

Line 362: was the sandy loam soil a natural soil?

Line 363: How many seeds were sown in each pot?

Line 365: A mean temperature of 33ºC is very high even for maize, can you explain this temperature? Which were the day/night mean temperatures?

Line 365, 378, 383, …: Write properly the Celsius degree symbol.

Line 366: Is 7.8 the proper sunlight length for July-October? I don´t think so

Lines 368-369: Why don´t authors use the same concentration units for both tested products? (SBE in % and GB in g/L)?

Line 373: Which was the day-interval between applications? Or maybe there was just one application? Please, explain as you say “foliage treatments”.

Line 377: accuracy of the balance?

Line 382 and 388: How many plants per pot were used for these 0.5 g? From just one plant? Or 2 plants?

Line 389: Please explain the meaning of EC0, EC1 and EC2

Line 402: Please write properly the following chemical formula: H2SO4 and more chemical formulas throughout the entire paper.

Line 460: Which were the ANOVA analyze factors?

Reviewer 4 Report

Review of the manuscript

Influence of Glycinebetaine (Natural and Synthetic) on Growth, Metabolites and Yield Production of Drought-Stressed Maize (Zea mays L.) Plants

The manuscript described the work done by the authors in two maize cultivars subjected to water stress using field capacity as a method to apply water stress. They used a natural source of GB and a synthetic one; however, the extraction and protocol composition data are missing.

The introduction needs to be more focused on maize. Only ten references from the 42 used in this section are papers on this crop, while 13 are base on others. Results are poorly presented; extensive work needs tables and figures (figure 1 axes legend is missing) and the discussion section.

Material and methods need more explanation from the water stress treatments to the quantifications methods of the different variables. Overall, this paper must undertake significant corrections.

Round 2

Reviewer 2 Report

Overall the well revised manuscript can be accepted as such for publishing in Plants. The Authors addressed all my comments.

Author Response

Thanks for the recommendation

Reviewer 3 Report

The paper has an important improvement made by authors. However there are still some issues to be solved:

1.- Line 63: Authors state that maize needs 50-800 mm and 120-150 days to complete its cycle but I think they shouldn´t be so categoric as there can be longer maize cycles. I would include somewhere the word "around" when talking about both ranges.

2.-Lines 108-109: There must be a missing word in this sentence (perhaps "that" between "increase" and "was", as the sentence has no sense in that way for me. Please check it.

3.- In all figures, the boxes of the legends should be increased in size in order to properly see the pattern style of their filling.

4.- The Celsius temperature symbol should be properly written (ºC)

Author Response

Reviewer #. 3

1.- Line 63: Authors state that maize needs 50-800 mm and 120-150 days to complete its cycle but I think they shouldn´t be so categoric as there can be longer maize cycles. I would include somewhere the word "around" when talking about both ranges.

Now, the word “around” has been added

2.-Lines 108-109: There must be a missing word in this sentence (perhaps "that" between "increase" and "was", as the sentence has no sense in that way for me. Please check it

Corrected

3.- In all figures, the boxes of the legends should be increased in size in order to properly see the pattern style of their filling.

Size of the legends in boxes has been increased now.

4.- The Celsius temperature symbol should be properly written (ºC)

Corrected

Reviewer 4 Report

The authors made substantial modifications to the manuscript and I think this version is ready to be published.

Author Response

Thanks